# Learning Continuous Morphological Trajectories via Latent Principal Curves

## Abstract

Inferring continuous morphological transformations from a collection of static shapes is an important, yet challenging problem across various domains. In the context of cellular biology, prevailing approaches reduce 3D shape collections to static reconstructions or hand-crafted descriptors, which fail to capture smooth, multidimensional transitions. We present **MorphCurveVAE**, a two-stage pipeline for constructing continuous morphological trajectories from sets of static, segmented 3D cell images. Stage 1 learns a smooth, compact latent manifold of volumetric morphologies using a multi-branch convolutional variational auto-encoder (VAE) that can encode multiple correlated substructures into disentangled subspaces. Stage 2 extracts a constrained, topologically-aware principal curve through the augmented latent space to produce directional and correlated trajectories of the structures' dynamics. To demonstrate our framework, we apply MorphCurveVAE to a large public dataset (Allen Institute WTC-11) of segmented volumetric cell and nucleus images throughout the mitosis cycle. Our results indicate high-quality reconstructions, low projection errors to the fitted principal curve, and biologically and visually plausible continuous animations. These results suggest MorphCurveVAE as a practical tool for downstream generative modelling of morphological trajectories, and a methodological contribution for learning robust, ordered trajectories directly from image-derived latent spaces.

## 1 Introduction

Modeling continuous morphological changes is a cornerstone of many scientific domains (developmental biology, materials science, cellular simulations, morphometrics) (Welker et al., 2022; Litwin et al., 2024; Campello et al., 2020). Large repositories of segmented 3D images provide rich snapshot datasets, but converting such collections into continuous trajectories is not straightforward (Viana et al., 2023; Ulman et al., 2017). Many prior pipelines focus on per-sample reconstruction or rely on hand-crafted shape descriptors (e.g., volume, sphericity, elongation) that do not naturally capture smooth, multi-dimensional transitions (Ruan et al., 2019; Kazhdan et al., 2003). Trajectory-inference methods from genomics highlight the importance of topology-aware ordering (Saelens et al., 2019), yet existing 3D generative shape models do not provide analogous capabilities for morphological datasets, leaving continuous transitions underexplored. Moreover, recent cell culture modeling work continues to often poorly handle or completely omit transitional events in shape dynamics (Dixon, 2025).

In this paper we propose a general pipeline with two modular components: (i) a multi-branch VAE for learning smooth latent embeddings of volumetric morphologies, and (ii) a supervised principal-curve extractor that enforces ordered traversal through phase centroids while regularizing geometric arc properties. The pipeline, illustrated in Figure 1, is intentionally general, however we evaluate our method on a large public dataset of live human iPS cells collected and released by the Allen Institute (WTC-11 dataset) which provides volumetric masks for nucleus and cell membranes along with hand annotations for mitotic phase on a subset of cells (Viana et al., 2023). These same components apply to any domain where multiple correlated structures morphologically change together (e.g., multi-modal medical imaging, evolving organoids, time-lapse morphological studies).

We emphasize the following properties of MORPHCURVEVAE: (i) **Generality:** the multi-branch encoders accept an arbitrary number of correlated channels; the stochastic perturbation model allows for sampling plausible sets of trajectories with adjustable levels of variance; (ii) **Constrainability:** the curve extraction supports hard equality constraints (forcing the curve through ordered centroids) and soft regularizing penalties (arc-length, curvature), enabling topology-aware fitting for loops and cycles; (iii) **Reproducibility:** training/validation splits, hyperparameter grids, and comprehensive modules are provided for reproducible evaluation on new datasets.

## 2 RELATED WORK

**Representation learning and generative modeling of morphology.** Learning representations of 3D morphology has been approached through voxel- and mesh-based convolutional models, implicit-function methods, and more recently diffusion-based generative families. Voxel-based VAEs remain attractive because they directly accommodate segmented masks, allow probabilistic decoding, and integrate smoothly with constrained geometry processing (Wu et al., 2015; Ruan et al., 2019). Implicit approaches such as DeepSDF offer compact, continuous representations but often require more specialized decoders (Park et al., 2019). Diffusion-based and other likelihood-free models can achieve state-of-the-art geometric fidelity, including for microscopy-derived morphologies, but demand larger datasets, heavier compute, and more complex conditioning to enforce constraints (Waibel, 2022; Chu et al., 2023; GenerativeShapes, 2024). Our choice of a voxel-based dual-branch VAE reflects a pragmatic trade-off: interpretable, extensible to multiple correlated branches, and constrainable for trajectory extraction.

**Trajectory inference and principal curves.** A wide array of trajectory-inference (TI) methods exist, from diffusion pseudotime (DPT) (Haghverdi et al., 2016) to reversed-graph embedding (Monocle 2) (Qiu et al., 2017) and cluster-then-curve approaches like Slingshot (Street et al., 2018). Benchmark studies highlight that these methods are highly sensitive to latent topology, with cyclic trajectories posing persistent challenges (Saelens et al., 2019). Critically for our work, most TI algorithms are not designed for image-derived morphological masks, and the cyclic nature of many biological processes exacerbates their limitations. Recent work in trajectory modeling has begun exploring shape-informed embeddings and topology-aware extensions of principal curves, reinforcing the need for supervised, geometry-regularized formulations like ours (Anon, 2025). These observations motivate our supervised, topology-aware principal-curve extraction, which incorporates ordered-state constraints and explicit geometric regularization to robustly recover various cyclic morphological trajectories.

**Applications and datasets.** Representation learning and generative modeling of cellular morphology is an active and growing field. Recent large curated datasets, such as WTC-11, now enable systematic evaluation of reconstruction accuracy, latent separability, and trajectory extraction (Viana et al., 2023). Additionally, recent literature highlights practical challenges in culture-scale modeling, such as maintaining cell identity across divisions or accounting for mitosis in lineage tracking (Dixon, 2025; Cao et al., 2025). Our pipeline is designed to complement this ecosystem by providing a constrainable, reproducible method for extracting smooth and biologically meaningful trajectories from image-derived latent spaces.

## 3 METHODS

The MORPHCURVEVAE pipeline, represented in Figure 1, consists of two core stages: (i) learning a compact latent manifold of volumetric morphologies with a multi-branch convolutional variational autoencoder (VAE), and (ii) extracting a supervised, topology-aware principal curve through an augmented latent space – enabling the decoding of continuous, stochastically-sampled trajectories back to volumetric animations.

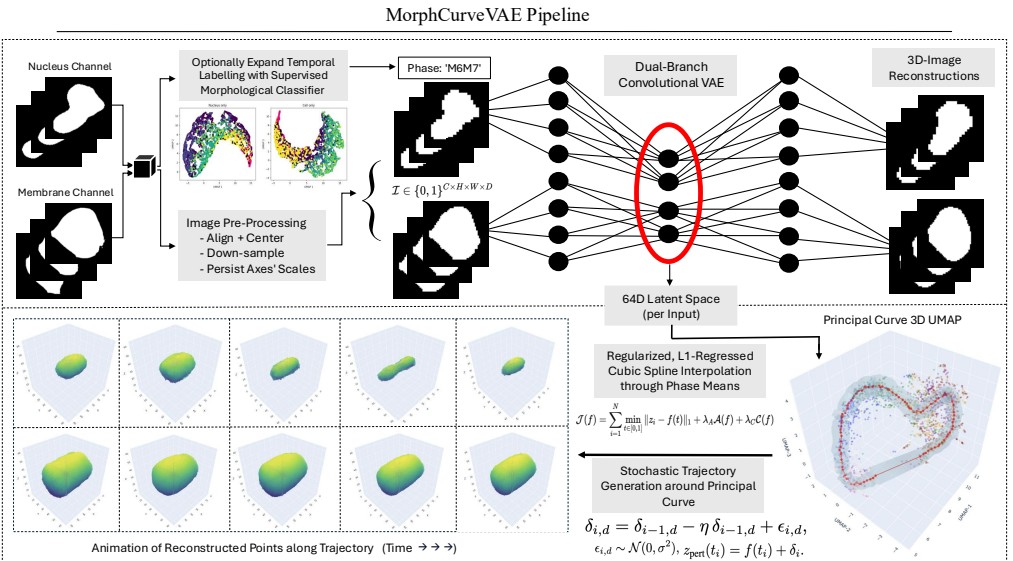

Figure 1: Pipeline overview: (top) multi-branch volumetric VAE encodes correlated channels into a compact latent space; (bottom right) supervised principal-curve extractor fits an ordered trajectory through the latent manifold; (bottom left) decoder projects stochastically sampled trajectory near principal curve back to shape space via the decoder, yielding reconstructed animation.

## 3.1 PREPROCESSING AND LABEL REFINEMENT

**Image preprocessing.** The pipeline accepts volumetric segmented masks (binary or probabilistic) for one or more correlated substructures. The minimal preprocessing performed by the pipeline includes (i) centering the masks and aligning the structures with ICP rigid alignment if needed, and (ii) downsampling images to a fixed grid of size $C \times D \times H \times W$ while persisting original scale axes for later augmentation (see Sec. 3.2.1). Note that for our dataset, images were previously both centered and aligned, and all inputs were downsampled to binary masks of size $2 \times 64 \times 64 \times 64$ with externally saved scale factors $(s_x, s_y, s_z)$.

**Label refinement and extension.** The principal-curve stage requires at least a subset of ordered temporal labels, and benefits greatly from a large set of high quality labels. When full annotation coverage is absent or mislabeling is suspected, we perform a light-weight label-refinement procedure prior to VAE training (details and code in Appendix A). In brief, we compute and utilize a morphological image signature for supervised classification both on the unlabeled data, and for internally assessing the quality of the given labeled data.

## 3.2 MULTI-BRANCH VARIATIONAL AUTOENCODER

**Architectural overview.** We adopt a multi-branch convolutional VAE that generalizes to an arbitrary number of correlated input channels. Each branch processes one input channel through a stack of 3D convolutional encoding blocks; branch embeddings are concatenated and then mapped to Gaussian latent parameters. The decoder mirrors the encoder and jointly reconstructs all channels.

**Configuration.** After an extensive grid-search of hyper-parameter candidate values using validation-based model selection (see Additional Materials), our instantiation of the VAE was then trained with the following specifications:

- **Input format:** $\mathcal{I} \in \{0,1\}^{2 \times 64 \times 64 \times 64}$ (channel 0: nucleus mask; channel 1: cell mask).
- **Encoders:** three encoding blocks, each consisting of Conv3D $\rightarrow$ GroupNorm $\rightarrow$ LeakyReLU $\rightarrow$ downsample (stride-2) with progressive channel growth (16, 32, 64 filters), were used in each branch.

- **Latent:** concatenated branch embeddings were mapped to Gaussian parameters $(\mu_\phi(x), \sigma_\phi^2(x)) \in \mathbb{R}^D$, $D = 64$; using an explicit 32/32 allocation for the two branches.

- **Decoder:** three transpose-convolutional blocks mirroring the encoder, outputting per-voxel logits for each channel, then converted to probabilities by a sigmoid before thresholding.

**Probabilistic model and loss.** The encoder defines an approximate posterior

$$q_\phi(z \mid x) = \mathcal{N}\big(z; \mu_\phi(x), \operatorname{diag}\big(\sigma_\phi^2(x)\big)\big), \tag{1}$$

and the decoder defines $p_\theta(x|z)$ producing per-voxel probabilities $\hat{x}_v$. We minimize the $\beta$-VAE evidence lower bound:

$$\mathcal{L}_\beta(\theta, \phi; x) = \mathbb{E}_{q_\phi(z|x)}\left[ -\sum_v \Big(x_v \log \hat{x}_v + (1 - x_v) \log(1 - \hat{x}_v)\Big)\right] + \beta \operatorname{KL}(q_\phi(z \mid x) \,\|\, p(z)), \tag{2}$$

with $p(z) = \mathcal{N}(0, I)$. The binary cross-entropy reconstruction term is suitable for binary masks. At inference we optionally threshold $\hat{x}_v$ at $\tau$ to obtain binary masks (default $\tau = 0.5$, however adjusting $\tau$ can improve visual quality in heavily imbalanced volumes).

### 3.2.1 LATENT SUBSPACE DESIGN AND SCALE AUGMENTATION

To preserve interpretable substructure we allocate contiguous sub-blocks of the latent vector to each branch (demo: 32 dims per branch). To preserve absolute size information lost by downsampling, we compute axis-wise scale factors $(s_x, s_y, s_z)$ for each sample and append them to the core latent $z$ to form an augmented latent

$$\tilde{z} = [z; s_x; s_y; s_z] \in \mathbb{R}^{D+3}. \tag{3}$$

During decoding we ignore the appended scale coordinates when passing to $p_\theta(\cdot \mid \cdot)$, and after decoding we rescale the reconstructed volume by $(s_x, s_y, s_z)$ to approximate original physical proportions. The appended scale coordinates are explicitly included in the principal-curve extraction and — in our implementation — are scaled to hold twice the variance of the latent space dimensions in order to be weighted more heavily, ensuring absolute size is considered in the trajectory geometry.

### 3.3 SUPERVISED, TOPOLOGY-AWARE PRINCIPAL-CURVE EXTRACTION

After training the VAE, we compute augmented latent vectors $\{\tilde{z}_i\}_{i=1}^N$ and extract a smooth one-dimensional trajectory $f : [0,1] \to \mathbb{R}^{D'}$ summarizing morphological progression. We begin by computing centroids of the ordered temporal phases,

$$\bar{z}_k = \frac{1}{|I_k|} \sum_{i \in I_k} \tilde{z}_i, \tag{4}$$

where $\mathcal{I}_k$ indexes samples in stage $k$. These centroids are enforced as hard equality constraints on the curve.

We represent the trajectory as a sequence of $M$ points in latent space, of which the phase centroids are fixed and the remaining are optimized. A cubic spline interpolant $f(t)$ is then fit through these points, yielding a continuous, closed curve. The optimization minimizes a composite objective,

$$\mathcal{J}(f) = L_{\mathrm{reg}}(f) + \lambda_A \mathcal{A}(f) + \lambda_C \mathcal{C}(f), \tag{5}$$

where the regression term

$$L_{\mathrm{reg}}(f) = \frac{1}{N} \sum_{i=1}^N \min_{t \in [0,1]} \|\tilde{z}_i - f(t)\|_1 \tag{6}$$

penalizes L1-distances to the nearest curve point, the arc-length penalty

$$\mathcal{A}(f) = \int_0^1 \|f'(t)\| \, dt \tag{7}$$

encourages compact parameterizations, and the curvature penalty

$$\mathcal{C}(f) = \int_0^1 \|f''(t)\|_2^2 \, dt \tag{8}$$

suppresses oscillations.

The hyperparameters $\lambda_A$, and $\lambda_C$ can be adjusted to balance curve fidelity and regularization. Higher values generally favor more compact and direct curves, resembling optimal transport transformations, whereas lower values allow the curve to explore indirect trajectories between phase centroids, which may be useful if the temporal resolution of discrete phase labels is coarse. In this work, values were selected based on visual validation of the resulting trajectories, though cell IDs across imaged phases would allow for more rigorous evaluation of how well resulting curves recapitulate complete single-cell progression across phases.

We optimize the free curve points directly with L-BFGS-B, then construct $f$ by cubic spline interpolation through all optimized and fixed points. Initialization is given by piecewise linear interpolation through the centroids, and cyclic closure is enforced by wrapping the final segment back to the first centroid. We found this procedure numerically stable and robust across seeds.

### 3.4 Stochastic trajectory sampling and decoding

**Smooth perturbation process.** To generate realistic variant trajectories, we augment the principal curve $f(t)$ with smooth stochastic deviations. Rather than sampling from independent Gaussians per stage, we simulate a mean-reverting perturbation process along each latent dimension. For a base curve discretized into $N$ points $\{f(t_i)\}_{i=1}^N$, we construct deviations recursively as

$$\delta_{i,d} = \delta_{i-1,d} - \eta \, \delta_{i-1,d} + \epsilon_{i,d}, \quad \epsilon_{i,d} \sim \mathcal{N}(0, \sigma^2), \tag{9}$$

for dimension $d$ and step $i$, where $\eta > 0$ controls the reversion strength toward the unperturbed curve and $\sigma$ controls the noise scale. The deviations $\delta_{i,d}$ therefore follow an Ornstein–Uhlenbeck-like process, yielding smooth trajectories that remain close to the original curve. The final sampled trajectory is then obtained by adding the deviations to the base curve:

$$z_{\text{pert}}(t_i) = f(t_i) + \delta_i. \tag{10}$$

This construction produces stochastic trajectories that capture variability while avoiding divergence from the learned manifold. The parameters $(\sigma, \eta)$ control the magnitude and smoothness of deviations, and were selected to yield trajectories visually consistent with the latent morphology progression.

**Decoding and rescaling.** To produce volumetric frames, we (i) strip appended scale coordinates from $z_{\text{pert}}(t)$, (ii) decode the core latent through the trained decoder $p_\theta(\cdot \mid z)$ to obtain per-voxel probabilities, (iii) optionally threshold at $\tau$ to obtain binary masks, and (iv) resample/rescale the decoded volume by the corresponding $(s_x, s_y, s_z)$ to approximate physical dimensions. Resampling is implemented via trilinear interpolation to preserve smooth geometry. By sweeping $t$ across $[0, 1]$ (and sampling multiple perturbations per $t$) we synthesize continuous animations that reflect both the mean trajectory and empirically observed phase-wise variability.

## 4 Data and applicability

To demonstrate the utility of the MORPHCURVEVAE pipeline, we applied it to a public collection (Allen Institute WTC-11; Viana et al. 2023) of segmented volumetric images containing two channels (nucleus and cell membrane). The image dataset also includes comprehensive hand annotations of mitotic phases (5,764 cells labeled across a multi-stage mitotic taxonomy). These labels reflect the sequential stages of mitosis, beginning with interphase ($M0$) and progressing through $M1M2$ (prophase–prometaphase), $M3$ (metaphase), $M4M5$ (anaphase–telophase), and $M6M7$ (cytokinesis, further split into early/half). More details are available from the Allen Institute (Allen Institute, 2023). Categories for non-mitotic or corrupted examples are also included: (`blob`, `wrong`, `dead`). Label counts are: $M0$=2516, $M1M2$=441, $M3$=823, $M4M5$=927, $M6M7\_early$=338, $M6M7\_half$=561, with 20 `blob`, 35 `wrong`, and 103 `dead`. To reduce potential learned imbalance in the VAE, phase label counts can optionally be balanced prior to training.

Although not required for our priorly centered and aligned images, the pipeline does support this vital processing step using ICP-alignment. All images were then downsampled to a fixed resolution of $2 \times 64 \times 64 \times 64$ (channel, depth, height, width). As downsampling removes absolute size information, we computed and stored an additional three scale factors (one for each physical axis).

Additionally, the pipeline requires at least a subset of labeled examples when input. To improve label quality, we trained a small probabilistic classifier using a light-weight, rotationally-invariant morphological representation (see Appendix A). Pre-annotated cells that were predicted with more than fivefold confidence in favor of a different phase were pruned from the training set, thereby reducing likely mislabels (discarded $\sim 6\%$ of samples). After this pruning step, the same classifier can be used to extend phase annotations to unlabeled cells if necessary, prior to balancing training splits across annotation bins.

These steps ensure that both the morphological inputs and phase labels are standardized, interpretable, and robust for subsequent representation learning and trajectory extraction. Datasets suitable for this pipeline should provide: (i) per-sample segmented masks for the correlated structures of interest (binary or probabilistic), (ii) a subset of ordinal annotations to supervise trajectory extraction.

## 5 EXPERIMENTS AND RESULTS

We evaluated the proposed dual-branch VAE pipeline on the Allen WTC-11 segmented 3D cell dataset. Our experiments focus on (i) hyperparameter optimization and model training, (ii) latent-space organization, (iii) reconstruction fidelity, (iv) principal curve trajectory extraction, and (v) stochastic trajectory generation.

### 5.1 HYPERPARAMETER SEARCH AND MODEL TRAINING

We performed a structured hyperparameter search to select the optimal VAE configuration. Data were randomly split into 80/20 train/validation folds, and models were trained using Adam, batch size 2, a ReduceLROnPlateau scheduler (halving the learning rate if validation loss plateaued), and early stopping with a patience of 2 epochs (maximum 20 epochs).

For each latent dimension considered ($D \in \{16, 32, 64\}$), candidate configurations were generated using a random grid sampling over learning rate, $\beta$ regularization, convolutional stride, and base channel size. The best-performing configuration per latent dimension was selected based on minimum validation reconstruction loss, with larger latent dimensions yielding progressively lower loss at the cost of latent-space compactness.

Table 1: Summary of selected hyperparameter configuration for Dual-Branch VAE (64 latent dims).

| Latent Dim | LR | $\beta$ | Stride | Base Channels | Training Epochs | Final Val Loss |
|---|---|---|---|---|---|---|
| 64 | 0.0005 | 0.2 | 2 | 16 | 10 | 16455.63 |

Complete per-run logs, including per-epoch validation losses and full candidate configurations, are provided in the Additional Materials.

### 5.2 LATENT-SPACE ORGANIZATION

To evaluate whether the latent embeddings captured biologically meaningful variation, we projected the 64D latent space to 2D using UMAP (McInnes et al., 2018). Figure 2 shows these embeddings colored by (i) mitotic phase, (ii) nuclear volume, (iii) height, and (iv) sphericity. The latent space observably reflects the substantial progression between the continuous temporal stages, while also demonstrating the correlation of different individual structural properties to each phase. These hand-crafted descriptors are insufficient to capture the extensive variability among morphologies, however they can offer valuable and interpretable insights.

Additionally, the separation of the mitotic phases is evidently more pronounced when examining the nuclear latent space rather than that of the cell-body. This observation aligns with biological expectations: the nucleus undergoes relatively discrete structural changes during mitosis—flattening, condensation, and division — whereas the cell membrane deforms more gradually and continuously and in response to environmental cues.

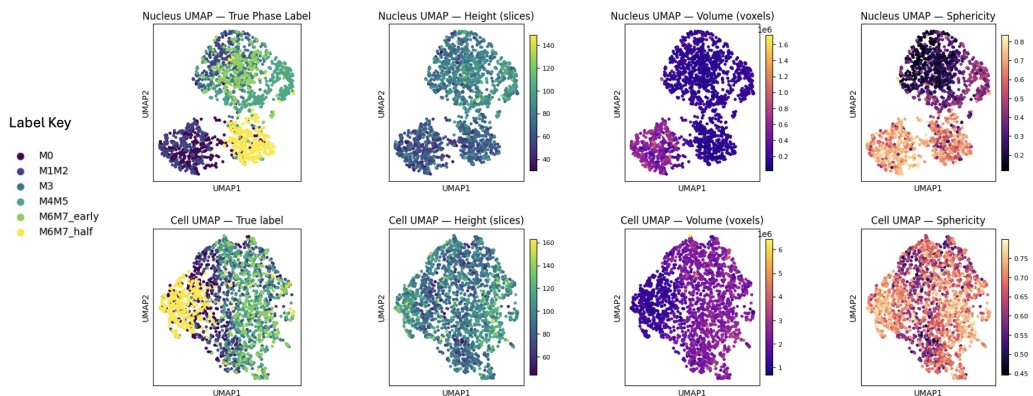

Figure 2: 2D UMAP projection of latent embeddings colored by (A) mitotic phase, (B) nuclear volume, (C) height, and (D) sphericity.

## 5.3 RECONSTRUCTION FIDELITY

We evaluated VAE reconstruction performance using both voxel-wise accuracy and Dice scores for foreground segmentation to mitigate background dominance. Table 2 summarizes these metrics per mitotic phase, demonstrating the VAE's robustness to diverse morphologies. Figure 3 shows central 2D slices of example reconstructions, demonstrating visually accurate recovery of morphological detail across all stages.

Table 2: Reconstruction performance (test set) by mitotic phase.

| Phase | Voxel Accuracy (%) | Voxel STD (%) | Dice Score (%) | Dice STD (%) |
|---|---|---|---|---|
| M0 | 98.38 | 0.48 | 88.99 | 3.08 |
| M1M2 | 97.99 | 0.48 | 88.41 | 3.26 |
| M3 | 98.54 | 0.32 | 91.99 | 1.87 |
| M4M5 | 98.70 | 0.32 | 92.99 | 1.57 |
| M6M7-early | 98.69 | 0.25 | 93.35 | 1.41 |
| M6M7-half | 98.81 | 0.26 | 92.16 | 1.81 |
| Overall | 98.52 | 0.46 | 91.29 | 2.98 |

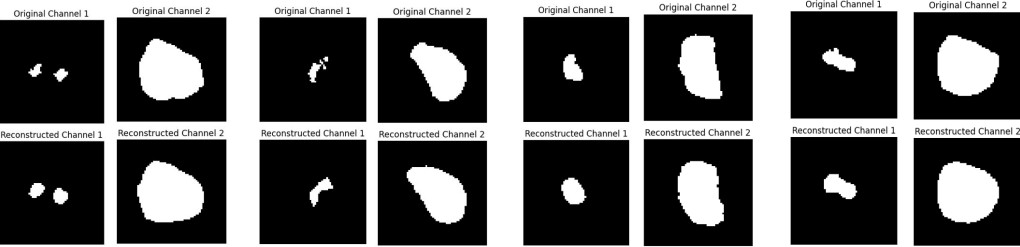

Figure 3: Central slices of several example reconstructions.

## 5.4 PRINCIPAL CURVE TRAJECTORY EXTRACTION

Within the 64-dimensional latent space, we fit a constrained principal curve through the cell embeddings, explicitly enforcing that the curve passes through the phase-wise centroids. This curve captures the dominant morphological progression across mitosis, though a one-dimensional trajectory cannot fully account for all latent variability.

To assess how well the curve represents the embedding, we computed projection distances from each latent point to the curve. Key quantiles are reported in Table 3, with distances falling between 0.27 and 0.47 times the mean latent norm (10th–90th percentiles), substantially tighter than would be expected for a random one-dimensional curve in high dimensionality, which typically lies at roughly the same scale as the full latent norm itself. This contrast indicates that the constrained principal curve successfully captures meaningful low-dimensional structure.

Table 3: Quantiles of latent projection distances to the constrained principal curve, measured in median vector norms.

| Quantile | 10% | 25% | 50% | 75% | 90% |
|---|---|---|---|---|---|
| Distance | 0.2713 | 0.32126 | 0.3793 | 0.4413 | 0.4747 |

Figure 5.4 shows a 3D UMAP projection of the latent space with cell embeddings colored by mitotic phase, overlaid with the principal curve. Individual points along the curve illustrate reconstructed morphologies, with animations available in the Additional Materials. A tubular envelope around the curve encodes local variance of projections, highlighting regions of higher morphological heterogeneity. Notably, a pronounced gap occurs in the latent space corresponding to cytokinesis, where segmentation captures only one daughter cell, producing a discrete jump within the otherwise smooth, cyclic trajectory. This visualization clearly conveys the principal morphological progression throughout mitosis and the overall cyclic nature captured by the model.

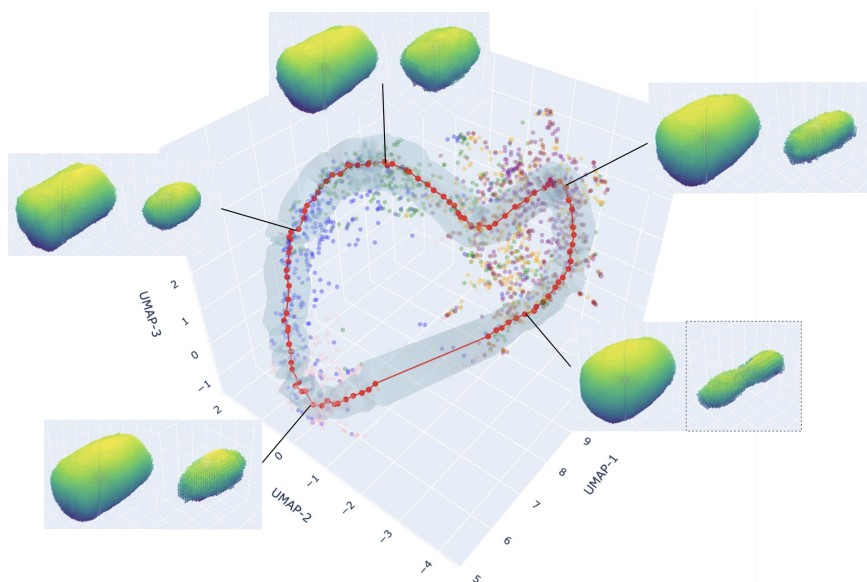

Figure 4: 3D UMAP projection of latent embeddings with the principal curve overlaid as a tube, where thickness represents local variance of projections computed via equal-length intervals and perpendicular distances.

## 6 CONCLUSION

We have presented MORPHCURVEVAE, a reproducible pipeline for transforming collections of static, segmented 3D images into continuous structural trajectories, enabling stochastic generation of plausible morphological populations. Our contribution addresses the lack of general-purpose tools for converting repositories of volumetric shapes into interpretable, morphological dynamics, thereby extending the ecosystem of generative and trajectory-based modeling methods.

Compared with state-of-the-art 3D generative models such as diffusion-based or implicit-function approaches (Waibel, 2022; Chu et al., 2023; GenerativeShapes, 2024), MORPHCURVEVAE prioritizes interpretability, constraint enforcement, and computational efficiency. While diffusion-based models can yield high-fidelity geometry, they often require larger datasets, heavier compute budgets, and do not natively support the constraints which enable our pipeline's topological robustness (OctFusion, 2024; Waibel, 2022). Our pipeline instead provides a lightweight and flexible option for annotated microscopy datasets where practical constraints and interpretability are paramount.

When evaluated on a relevant dataset, MORPHCURVEVAE produced high-fidelity reconstructions, pronounced latent progression between phases, and visually coherent animations, demonstrating both robustness and practical utility. In summary, MORPHCURVEVAE provides a principled framework for bridging static morphological datasets with temporally structured generative modeling, offering a foundation for future advances in dynamic shape analysis.

Several limitations of the pipeline, as well as its application to the dataset, should be noted. Firstly, our design choice to use voxel-based VAEs incurs a trade-off of weaker geometric fidelity compared to signed-distance or diffusion-based generators, for the sake of interpretability and constrainable flexibility. Pertaining to the dataset (Allen Institute WTC-11), no persisting image IDs exist across temporal phases, obstructing attempts to benchmark accuracies for complete reconstructed trajectories of individual structures. This dataset also contains only single-cell segmentations; thus, the trajectories do not show the physical splitting into two daughter cells but rather collapse onto one of them after the split, obfuscating the exact dynamics. Lastly, any discrete labeling for intrinsically continuous trajectories are noisy and approximate, thus constraining the curve to pass through discrete centroids is imprecise.

Future directions of investigation include: (i) experimenting with higher-fidelity encoding methods, (ii) accommodating dividing trajectories, and (iii) developing parallel methodologies requiring less supervision.

## 7 REPRODUCIBILITY STATEMENT

Please see the code repository referenced below:

- **Code Repository:** `https://github.com/anonymous67552/MorphCurveVAE`
- **Additional Materials:** Hyperparameter grid-search logs, extended figures and animations: `https://github.com/anonymous67552/AdditionalMaterials`

The associated README file contains all necessary instructions to reproduce these results.

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

## A APPENDIX: ROTATIONALLY-INVARIANT MORPHOLOGICAL EMBEDDINGS FOR LABEL REFINEMENT

To ensure balanced training splits across mitotic phases, the pipeline requires reliable phase annotations prior to autoencoder training. Because the latent space is not yet available at this stage, we construct a rotationally invariant morphological embedding directly from the original volumetric masks. Specifically, surface points are sampled from each binary mask, and all pairwise Wasserstein-like distances between points are computed. From the resulting distance distribution, 100 quantiles are extracted to summarize shape geometry in a way that is robust to outliers. This procedure is performed independently for the nucleus and cell body, after which the two 100-dimensional summaries are concatenated to yield a 200-dimensional embedding vector for each cell. While this embedding has no decoder and cannot reconstruct shapes, it provides a compact, rotationally invariant signature of the morphology. Empirically, these embeddings exhibit meaningful separability across phases in low-dimensional UMAP projections (available in Additional Materials).

We then train a supervised classifier on these embeddings using available phase annotations. The classifier is implemented as a stacked ensemble of diverse base learners—random forests, $k$-nearest neighbors, support vector machines, and gradient-boosted trees—with a random forest meta-learner.

Training is performed in a two-fold procedure (50/50 split), where the ensemble is trained on one half of the labeled data and used to predict the other. Any cell for which the classifier predicted a phase different from the annotation with more than fivefold higher confidence was flagged as a probable mislabel. These samples (approximately 6% of annotated cells) were excluded from training, reducing label noise while retaining the majority of the dataset. After pruning, the same classifier can be applied to extend phase annotations across the entire dataset when desired.

Despite the intrinsic ambiguity of phase boundaries, the ensemble achieved over 90% classification accuracy even without excluding mislabeled or low-quality images (e.g., cells annotated as "dead," "blob," or "wrong"). This indicates that the rotationally invariant embedding provides a reliable morphological signature for phase classification and that the pruning criterion effectively filters a small fraction of likely erroneous labels, yielding cleaner supervision for downstream representation learning.

