# OpenReview forum: "Learning Continuous Morphological Trajectories via Latent Principal Curves"
_ICLR.cc/2026/Conference — ICLR 2026 Conference Withdrawn Submission_

### Official Review · Reviewer_XGER · 2025-10-26

**Soundness:** 2
**Presentation:** 3
**Contribution:** 2
**Rating:** 2
**Confidence:** 4

**Summary:**

The paper proposes a latent embedding of 3D voxelized cell masks and extracts a continuous temporal trajectory that mimics the cell's morphological evolution. There are several heuristic choices, such as augmentation and curve fitting in the latent dimension, which the reviewer assumes largely stem from the limited data size. The approach demonstrates a good performance in a small set of available data.

**Strengths:**

The approach is carefully designed to perform well in the limited dataset and adapts some popule techniques to the problem setting.

**Weaknesses:**

The authors use various technical components of 3D geometry processing but appear to lack a clear understanding of some of them.

- One thing that troubles me reading the paper is that the authors often describe implicit-function and diffusion-based methods in parallel as something recent but not applicable to the problem setting. An implicit function concerns data representation and may take more time to extract the explicit shape, but does not necessarily require more data to train. And in my opinion, if designed cleverly, it can still replace the low-resolution voxel representation and provide a continuous, smooth volume model within the current framework. On the other hand, diffusion-based methods can process both explicit and implicit representations, and the model structure falls into a different category. It is true that it requires more data to train, but there are trade-offs as well. The arguments related to implicit or diffusion appear rather short-sighted, lacking a comprehensive understanding of the relevant fields.

- Figure 1 is too small, and uses various jargons without much explanation. For example, a phase can mean different things and needs to be clearly defined before use. The figure is also misleading as it shows 2D masks, but they are actually 3D masks.

- The scaling in the work does not look appropriate and needs to be further verified. I've never seen such a formulation— augmentation and re-scaling while using a separate portion of the latent vector. The entire process appears very unique.

- The label refinement needs more explanation.

- The biggest concern about the framework is that the VAE only enforces the reconstruction of input data and a uniform IID Gaussian in the latent space, and does not consider temporal smoothness within the latent space. However, the work assumes that a principal curve can correctly follow the temporal trajectory. While the results appear reasonable for the presented method, it would be more appropriate to enforce necessary prior conditions, such as Lipschitz constraints. The centorid constraint (Equation (4)) may serve as a heuristic to overcome the limitations of the current VAE, which is rather hard-coded and may conflict with the embedding space without proper regularization during latent embedding learning.

**Questions:**

- Which cubic spline are you using? As far as I understand, cubic splines are piece-wise polynomials and have four control points for each section. How do you subdivide the entire curve into a set of cubic splines?

- Figure 2 is also too small and should be edited so that it can be fully interpreted at 100%. When zoomed in, the second column is "height" and the third column is "nuclear volume", which is the order swapped from that in the caption. Also, please add (A), (B), (C), and (D) in the Figures as well.

- What are dice scores?

- Table 3 may not contain much information.

- In line 401, Figure 5.4 -> Figure 4

- It is very hard to understand the thickness in Figure 4 or notice any variation in thickness in the figure. Please revise.

- What do you mean by topological robustness in line 443?

- What do you mean by line 462-463: developing parallel methodologies requiring less supervision. Can you elaborate more?

---

### Official Review · Reviewer_DiwQ · 2025-10-28

**Soundness:** 2
**Presentation:** 2
**Contribution:** 1
**Rating:** 4
**Confidence:** 3

**Summary:**

This paper introduces a pipeline named MorphCurveVAE, which aims to learn latent representations and continuous morphological trajectories from 3D cell images. The proposed pipeline consists of a dual-branch VAE for learning latent representations from input segmentation masks, and a trajectory generation module. The VAE component adopts the standard β-VAE framework and extends it through image pre-processing and multi-branch information fusion. The trajectory generation module optimizes a composite objective that includes a curve-fitting term and topological constraints. The pipeline is evaluated on the public WTC-11 cell dataset. The paper presents qualitative and quantitative results showing that the multi-branch VAE learns meaningful latent embeddings and achieves good reconstruction fidelity. The trajectory generation appears reliable, as evidenced by the smaller projection distance compared with random curve baselines.

**Strengths:**

1. The research problem of morphological trajectory generation is interesting and clearly defined.
2. The use of a VAE framework for latent representation learning and trajectory curve generation is reasonable and technically sound.
3. The authors provide the code and detailed experimental records, which enhance the reproducibility and transparency of the work.

**Weaknesses:**

1. The technical contribution appears limited, as the proposed pipeline largely follows standard VAE modeling and interpolation strategies. While it may serve as a solid biomedical application study, it may not meet the level of methodological innovation typically expected at ICLR. The paper provides limited new insights into the underlying problem.
2. Despite that the paper shows smooth and reasonable morphological trajectories, it does not provide any biological or quantitative evidence that these trajectories reflect real or meaningful cellular changes, nor does it explain how the results could actually help biologists.
3. The experiments are conducted on only one dataset, which weakens the evidence for the generalization and robustness of the proposed approach. In addition, although the paper mentions signed-distance and diffusion-based methods as related work, none of these existing methods are included for comparison.
4. The reconstruction results appear overly smooth given that the inputs are simple binary masks. The authors could consider VAE/MAE-GAN to improve reconstruction fidelity.
5. Section 3.3 mentions that “These centroids are enforced as hard equality constraints on the curve,” but the specific formulation of this constraint is not provided. Some other descriptions are also ambiguous or insufficiently detailed (see Questions).

**Questions:**

1. Do the different VAE branches share any weights, or are they trained independently?
2. How did you implement $f(t)$ defined in Section 3.3? Is it parametric or non-parametric?
3. For the penalization terms $\mathcal{A}$ and $\mathcal{C}$, are these original contributions of this work or adopted from existing literature?
3. How densely did you sample $t$ in actual calculation? This choice likely affects the numerical stability of computing $f^{'}(t)$ and $f^{''}(t)$.
4. What is the rationale for using the L1-norm instead of the more conventional L2-norm for regression in Eq. (6)?
5. For the WTC-11 dataset, did you create a separate test split for all evaluations (e.g., results in Table 2, Table 3, and Figure 4)?

---

### Official Review · Reviewer_64fj · 2025-10-31

**Soundness:** 3
**Presentation:** 2
**Contribution:** 2
**Rating:** 2
**Confidence:** 3

**Summary:**

The paper tackles a challenging and understudied problem in 3D cellular morphology - namely, how to infer continuous morphological transitions from collections of static volumetric data. This represents an important direction for bridging generative modelling and cell biology.

**Strengths:**

The paper tackles a challenging and understudied problem in 3D cellular morphology - namely, how to infer continuous morphological transitions from collections of static volumetric data. This represents an important direction for bridging generative modelling and cell biology.

**Weaknesses:**

- The motivation for inferring morphological trajectories rather than studying static representations is not clearly articulated. While the idea of modelling continuous transitions is intuitively appealing, the paper does not convincingly argue why this temporal interpolation is biologically necessary or what specific insights it yields beyond static latent representations in 3D. The introduction would benefit from a clearer justification of the biological or analytical value of trajectory inference in this context.
- The discussion of prior literature on morphological representation learning is limited. Several recent methods have explored voxel-, mesh-, and implicit-function-based learning of 3D cellular morphology, as well as diffusion and generative models that achieve higher geometric fidelity. These should be more comprehensively reviewed and contextualised to clarify how MorphCurveVAE advances the state of the art.
- The description of the multi-branch VAE is too concise to fully understand the information flow between the nucleus and cell branches. It is unclear whether each biological compartment is reconstructed independently or if there is a mechanism for learning shared or correlated structure between them. The manuscript should clarify whether the reconstruction and regularisation losses are computed per branch or jointly across all channels.
- The quantitative evaluation of reconstruction quality relies primarily on voxel accuracy and Dice score. While these metrics assess segmentation fidelity, they do not capture perceptual or distributional similarity between reconstructed and real shapes. Incorporating established 3D generative metrics such as Fréchet Inception Distance (FID) or Maximum Mean Discrepancy (MMD) would provide a more robust and comparable assessment of generative performance.
- Figure 3, which shows only central 2D slices of 3D reconstructions, does not effectively convey the volumetric fidelity of the model outputs. Including orthogonal or multi-view renderings would more holistically illustrate reconstruction quality.
- Although the paper emphasises trajectory inference from static inputs as a central contribution, there is no actual direct quantitative or qualitative assessment of this capability. I could be misled by the results of Table 2, but it remains unclear how accurately the model can infer or reconstruct a continuous morphological trajectory from a single static input, or how robust such inferred trajectories are across samples.

**Questions:**

- What is a ‘heavily imbalanced’ volume? (Defined after Equation (2)).
- What is ‘interpretable structure’ in the latent space defined in 3.2.1?

---

### Note · Authors · 2025-11-22

**Comment:**

Upon the provided feedback in the official reviews, we will be withdrawing this paper from consideration for ICLR 2026. Thank you for the thoughtful comments.

**Withdrawal Confirmation:**

I have read and agree with the venue's withdrawal policy on behalf of myself and my co-authors.